# Diet of Adult Sardine *Sardina pilchardus* in the Gulf of Trieste, Northern Adriatic Sea

Diego Borme [1] , Sara Legovini [2], Alessandra de Olazabal [1] and Valentina Tirelli [1,*]

[1] National Institute of Oceanography and Applied Geophysics—OGS, 34151 Trieste, Italy; dborme@ogs.it (D.B.); adeolazabal@ogs.it (A.d.O.)
[2] KCS Caregiver Cooperativa Sociale, Via A. Gramsci 7, 34074 Monfalcone, Italy; sara.legovini@libero.it
* Correspondence: vtirelli@ogs.it

**Abstract:** Food availability is thought to exert a bottom-up control on the population dynamics of small pelagic fish; therefore, studies on trophic ecology are essential to improve their management. *Sardina pilchardus* is one of the most important commercial species in the Adriatic Sea, yet there is little information on its diet in this area. Adult sardines were caught in the Gulf of Trieste (northern Adriatic) from spring 2006 to winter 2007. Experimental catches conducted over 24-h cycles in May, June and July showed that the sardines foraged mainly in the late afternoon. A total of 96 adult sardines were analysed: the number of prey varied from a minimum of 305 to a maximum of 3318 prey/stomach, with an overall mean of 1259 ± 884 prey/stomach. Prey items were identified to the lowest possible taxonomical level, counted and measured at the stereo-microscope. Overall, sardines fed on a wide range of planktonic organisms (87 prey items from 17 μm to 18.4 mm were identified), with copepods being the most abundant prey (56%) and phytoplankton never exceeding 10% of the prey. Copepods of the Clauso-Paracalanidae group and of the genus *Oncaea* were by far the most important prey. The carbon content of prey items was indirectly estimated from prey dry mass or body volume. Almost all carbon uptake relied on a few groups of zooplankton. Ivlev's selectivity index showed that sardines positively selected small preys (small copepods < 1 mm size), but also larger preys (such as teleost eggs, decapod larvae and chaetognaths), confirming their adaptive feeding capacity.

**Keywords:** *Sardina pilchardus*; small pelagic fish; diet composition; dietary carbon; feeding cycle; feeding selectivity; Mediterranean Sea

## 1. Introduction

Small pelagic fish play a crucial role in marine food webs, as key species in energy transfer from plankton to larger predators, marine mammals and seabirds [1,2]. Short lifespans, high fecundity rates and planktivorous feeding behaviour make small pelagic fishes sensitive to changing oceanographic conditions [3]. However, the ability of anchovies and sardines to adapt to different trophic conditions and the plasticity of their feeding behaviour have often been linked to their "ecological success" [4–6].

Fish exploitation and environmental changes may exert an important combined effect on small pelagic fish stocks. In the Mediterranean Sea, several authors have supported the hypothesis that food availability and/or trophic competition may exert a bottom-up control on the growth, size and body condition of small pelagic fish [7,8]. Therefore, studies on the trophic ecology of small pelagic fishes represent fundamental knowledge to improve the assessment and management of these species.

Previous studies based on numerical dietary indices described sardines as exclusively or almost exclusively phytophagous fishes. This was especially true for sardines living in upwelling areas, where it is commonly assumed that high phytoplankton production supports short and efficient food chains [9–11]. However, this assumption has been challenged following the assessment of carbon uptake by different food items [4,12]. It has

been shown that the diets of both sardines and anchovies rely significantly more heavily on zooplankton than on the more numerous cells of phytoplankton for carbon content [12].

There is great intraspecific variability in the morphology of feeding structures between sardine populations from different regions [13]. Sardines live in both highly productive upwelling systems [4,11] and more oligotrophic waters [14–17], probably thanks to their ability to switch to the most appropriate feeding mode to optimise energy intake from the trophic environment. In particular, sardines from the Mediterranean have fewer gill rakers and greater spacing than those from the Atlantic [15]. This difference has been explained as an adaptation to the higher availability of phytoplankton in the Atlantic, where sardines have an advantage due to their filtration behaviour. In the Mediterranean Sea, the diet of sardines was mainly investigated in the Gulf of Lion [5,15,18–20] and in the Aegean Sea [16,21–23], but little information is available for the Adriatic Sea, where only a few studies have been conducted on the diet of adult sardines in Croatian waters (central–eastern Adriatic) [24–27].

The European sardine *Sardina pilchardus* (Walbaum, 1792) is one of the most important commercially exploited species in the Mediterranean Sea, where it is caught by purse seine fishing with artificial lighting and pelagic trawling. The northern Adriatic is one of the most productive areas of the Mediterranean [28,29], and in the entire basin the average sardine landing from 1975 to 2013 was 45,000 t/year [30], which, together with anchovy, represents over 97% of the small pelagic catches. Landings of small pelagic species in the Adriatic Sea were estimated to be around 74 million (2013), representing almost one fifth of the total fish production in this Sea. Nevertheless, landings of small pelagic fish in the Mediterranean Sea have been characterised by a general downward trend in recent years [31].

The aim of this study is to improve the knowledge of the feeding habits of adult *S. pilchardus* in the northernmost part of the Adriatic Sea (Gulf of Trieste), and in particular: (i) to describe the diel feeding cycle, (ii) to analyse the diet composition across different seasons, (iii) to assess carbon uptake and (iv) to estimate feeding selectivity. The results of this study will provide important information for modelling studies and contribute to a better understanding of the functioning of the pelagic food web in the northern Adriatic.

## 2. Materials and Methods

### 2.1. Study Area

The Gulf of Trieste (Figure 1) is the northernmost gulf of the Adriatic Sea, with an average depth of 20 m (maximum depth: 25 m), a surface area of 600 km$^2$ and a volume of 9.5 km$^3$. This semi-enclosed continental shelf area exhibits large oscillations of temperature (4–29.2 °C) and salinity (10–38.5) [32]. Termohaline stratification occurs during spring and lasts to autumn, while, during winter, the water column is generally homogenized by wind mixing [33]. The principal freshwater input comes from the Isonzo River discharges (mean flow of 82 m$^3$ s$^{-1}$), which contribute to about 90% of the freshwater inputs [34].

### 2.2. Field Sampling

Sardines were collected from May 2006 to February 2007 in the Gulf of Trieste (northeastern Adriatic Sea) (Figure 1). Sampling took place monthly and/or bimonthly and during each sampling day 6 to 8 consecutive tows, at least 2 h from each other, were performed to describe the feeding diel cycle. No sardines were found in samplings carried out in December and February (Table 1). In May, June and July 2006, fish were sampled with 1 or 2 monofilament gillnets, each 4 m high and 400 m long, with a knot-to-knot mesh size of 15.5 mm. In September, October, December 2006 and February 2007, fish were sampled with a semipelagic trawl net equipped with a 12 mm (knot-to-knot) mesh cod-end, towed at an approximate speed of 3.0 knots. Both fishing gears worked in the same water layer, being deployed from the bottom to a height of 4 m above the bottom. Setting and towing times were adapted to seasonal and environmental conditions to maximize the fishing success. Fish from each tow were sorted, immediately frozen on board at −20 °C to stop digestive processes and preserved at the same temperature until laboratory analysis.

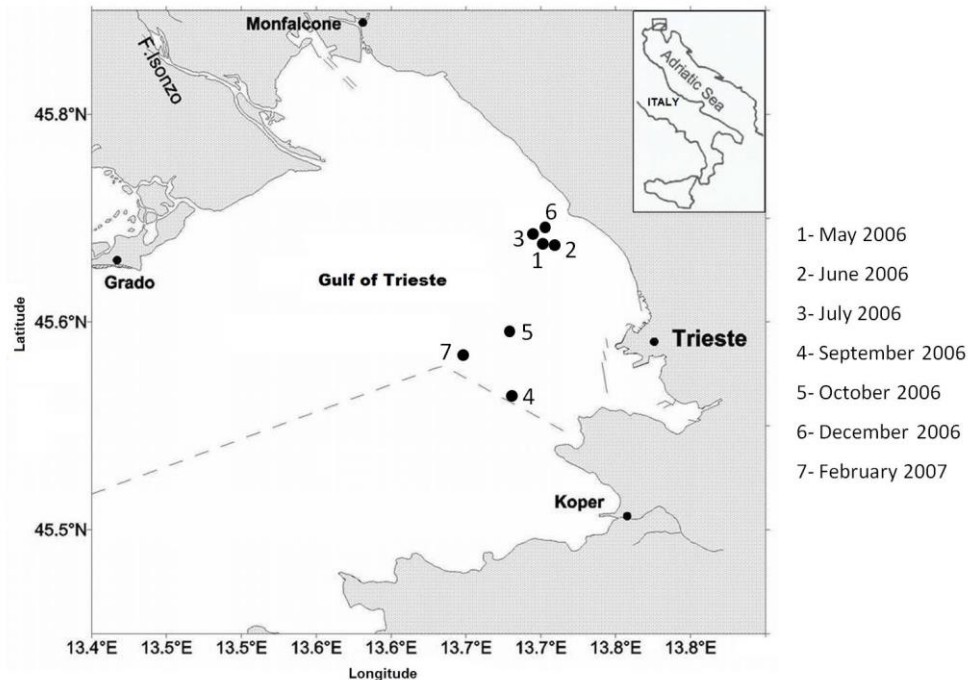

**Figure 1.** Map of the study area in the Gulf of Trieste (northeastern Adriatic Sea), with the ending positions of the fishing tows (●) (black dots) used to describe both sardine diet and mesozooplankton in the field.

**Table 1.** Sampling information: date; sunrise and sunset time at each sampling day; fishing gear used; number of performed tows during the 24 h cycle of sampling and in brackets the number of tows with at least 20 sardines. Time is expressed as (GMT + 1).

| Date | Sunrise | Sunset | Fishing Gear | Tows | Time of Tow Used for Diet |
|---|---|---|---|---|---|
| 10–11 May 2006 | 04:42 | 19:21 | gill net | 6 (4) | 20:50 |
| 20–21 June 2006 | 04:15 | 19:57 | gill net | 8 (7) | 20:40 |
| 25–26 July 2006 | 04:40 | 19:42 | gill net | 6 (6) | 18:15 |
| 04–05 September 2006 | 05:28 | 18:40 | trawling | 8 (2) | - |
| 26–27 October 2006 | 06:34 | 17:03 | trawling | 7 (3) | 17:05 |
| 14 December 2006 | 07:36 | 16:21 | trawling | 8 (0) | - |
| 01 February 2007 | 07:28 | 17:09 | trawling | 6 (0) | - |

Zooplankton samples were collected during fishing operations, generally after the retrieval of the fishing net. Vertical plankton tows from about 3 m from the bottom up to the surface were performed with a standard WP2 net (mesh size 200 μm; mouth opening diameter 58 cm). Immediately on retrieval of the net, plankton samples were fixed and preserved in a seawater-buffered formaldehyde solution (4% final concentration).

Temperature profiles were measured using a PNF-300 Profiling Natural Fluorometer probe in May, June and July 2006 and a portable thermometer YSI85 probe in September, October, December 2006 and February 2007.

### 2.3. Diel Feeding Cycle

At laboratory fish were defrosted, measured to the nearest 1 mm of total length (TL) and weighed on an analytical balance to the nearest 0.0001 g of total wet body mass. A total of 343 sardines (Table 2) were dissected and their stomachs were removed and preserved individually in a buffered 4% formaldehyde–seawater solution. Afterwards the stomachs were dissected and their contents were washed with distilled water on a Petri dish under a stereo-microscope. Subsequently, the stomach contents were filtered through pre-dried

and pre-weighed glass microfiber filters (Whatman® grade GF/F, 25 mm Ø) and dried at 60 °C until constant mass. To determine the diel feeding periodicity of sardine, the Fullness index (F) was calculated as

$$F = (DM/SWM) \times 1000 \text{ g SWM}$$

where DM is the dry mass of the stomach content and SWM is the somatic wet mass of fish. Feeding periodicity was described by plotting mean Fullness index (F) calculated in fish from the same tow against the time of day.

**Table 2.** Information about fish considered for diet composition analysis and for feeding cycle: number (n), total length $\pm$ st. dev. (TL), total wet mass $\pm$ st. dev. (TWM).

| | Diet Composition | | | Feeding Cycle | | |
|---|---|---|---|---|---|---|
| **Day** | **n** | **TL (mm)** | **TWM (g)** | **n** | **TL (mm)** | **TWM (g)** |
| 10 May 2006 | 24 | 173.83 $\pm$ 7.34 | 42.22 $\pm$ 4.76 | 81 | 175.56 $\pm$ 6.73 | 44.62 $\pm$ 4.98 |
| 20 June 2006 | 24 | 172.42 $\pm$ 5.32 | 45.09 $\pm$ 3.43 | 142 | 172.27 $\pm$ 8.97 | 44.38 $\pm$ 5.66 |
| 26 July 2006 | 24 | 174.08 $\pm$ 11.27 | 45.28 $\pm$ 6.66 | 120 | 174.97 $\pm$ 11.96 | 45.33 $\pm$ 7.02 |
| 26 October 2006 | 24 | 157.67 $\pm$ 15.17 | 33.11 $\pm$ 9.47 | - | - | - |

*2.4. Diet Composition and Dietary Carbon*

Only sardine specimens caught during the period of maximum feeding activity were analysed to describe the diet composition since prey were less digested and easier to identify. Digestive tracts' dissection took place under a stereo-microscope and the stomach content of each fish was washed out onto a Petri dish. All the material contained both in the cardiac stomach and in the *fundulus* of the stomach was considered as "stomach content". Regurgitation during sampling was not observed since no food was found in any oesophagus.

For each sampling period (Table 2), sardines were randomly divided in 3 groups of 8 individuals. The stomach contents of the 8 sardines were pooled and diluted in a known volume of 0.22 µm filtered seawater. After homogenization, subsamples representing one fish (1/8 of the original pool), were analysed under the stereo-microscope (Leica M205-C, up to 160× magnification). This procedure was repeated for 3 pools of 8 sardines each, in order to produce replicates. Prey items were identified to the lowest possible taxonomical level and counted. When specific characters were missing or damaged, copepod specimens of the genera *Paracalanus*, *Ctenocalanus*, *Clausocalanus* and *Pseudocalanus* were classified as the "Clauso-Paracalanidae" group. The prosoma length of all copepods and the maximum dimension of each other prey were measured using an ocular micrometer (accuracy of 6 µm). The original size of incomplete prey was reconstructed by means of morphometric relationship, obtained by the measurements of whole individuals captured in zooplankton samples. Prey size distribution was represented by grouping sizes in classes of 50 µm.

The carbon content of prey items was indirectly estimated applying relationship between dry mass or body volume and carbon content (Table A1).

*2.5. Feeding Selectivity*

Feeding selectivity was estimated by Ivlev's electivity index E [34], calculated as follows

$$E = (r_i - a_i)/(r_i + a_i) \tag{1}$$

where $r_i$ is the relative abundance of prey category i in the stomachs of fish (as a percentage of the total stomach contents) and $a_i$ is the relative abundance of the same prey at sea. E ranges from $-1$ to $+1$; negative and positive values indicating avoidance or positive selection for a prey category, respectively, and zero value indicating neutral selectivity.

Mesozooplankton samples collected in concomitance with fishing operations, were considered to define the food availability at sea. Taxonomic composition was analysed in subsamples sufficient to count and identify at least 1000 specimens. Mesozooplankton abundance was expressed as number of individuals per cubic meter of seawater. The volume of filtered water for each sample was estimated by multiplying the net-mouth area by the sampling depth.

## 3. Results

### 3.1. Temperature and Mesozooplankton in the Field

Sea surface temperature ranged from 10.1 °C to 26.1 °C, recorded in February 2007 and July 2006, respectively. From late spring to the beginning of autumn, a thermocline formed (stratified periods), while in October, December and February the seawater temperature was homogenous from the surface to the bottom (mixing periods) (Figure 2a). Salinity ranged from 36.9 to 38.2, presenting higher values in February. In May and June, the values were constant along the water column. In July, September and October, an alocline formed at a depth of 13–14 m. In December and February, salinity again showed constant values along the water column from the bottom to a depth of 2–3 m (Figure 2b).

Mesozooplankton composition and abundance were analysed in samples collected in correspondence to the fishing hauls dedicated to fish diet analysis. Mesozooplankton abundance ranged from 2044 to 13,212 ind. m$^{-3}$, measured in December and July, respectively (Figure 3). Copepods were generally the most abundant group with the exception of July and September when Cladocerans numerically dominated (6961 and 6352 ind. m$^{-3}$, respectively). Copepods were mainly represented by the order Calanoida from February to September (72–91% of Copepods) and by the Cyclopoida of the suborder Ergasilida in October and December (54% and 42%, respectively). Cnidarians were particularly abundant in June (2404 ind. m$^{-3}$), mainly with *Muggiaea* spp. and other Siphonophorans. Meroplankton, composed essentially of Echinoderms and Molluscs, showed its maximum (up to 2525 ind. m$^{-3}$) in June and July (Figure 3). Chaetognaths' abundance ranged from the minimum of 4 to the maximum of 78 individuals m$^{-3}$ in May and December, respectively. Detailed information is available in the Table A2.

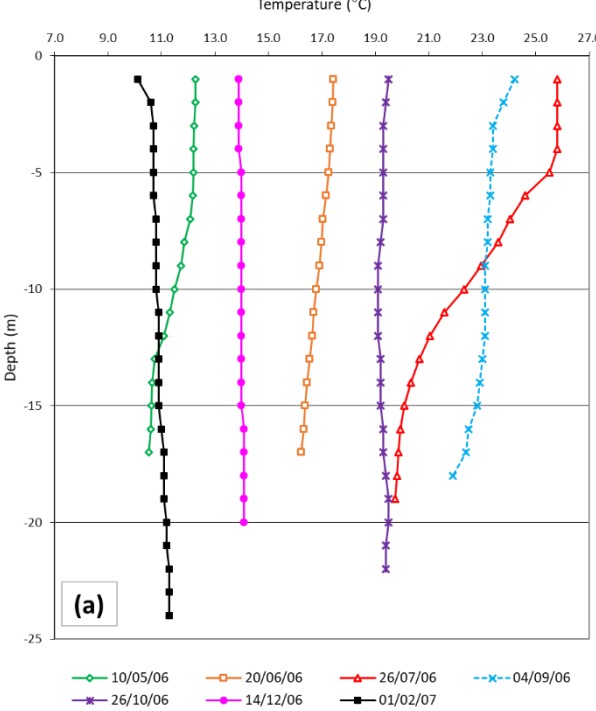

**Figure 2.** *Cont.*

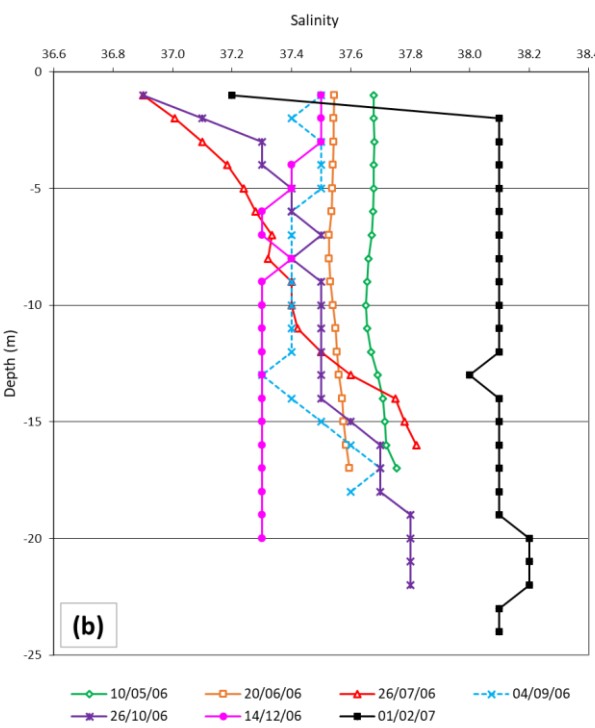

**Figure 2.** Temperature (**a**) and salinity (**b**) profiles measured close to positions where sardine diet and mesozooplankton were described.

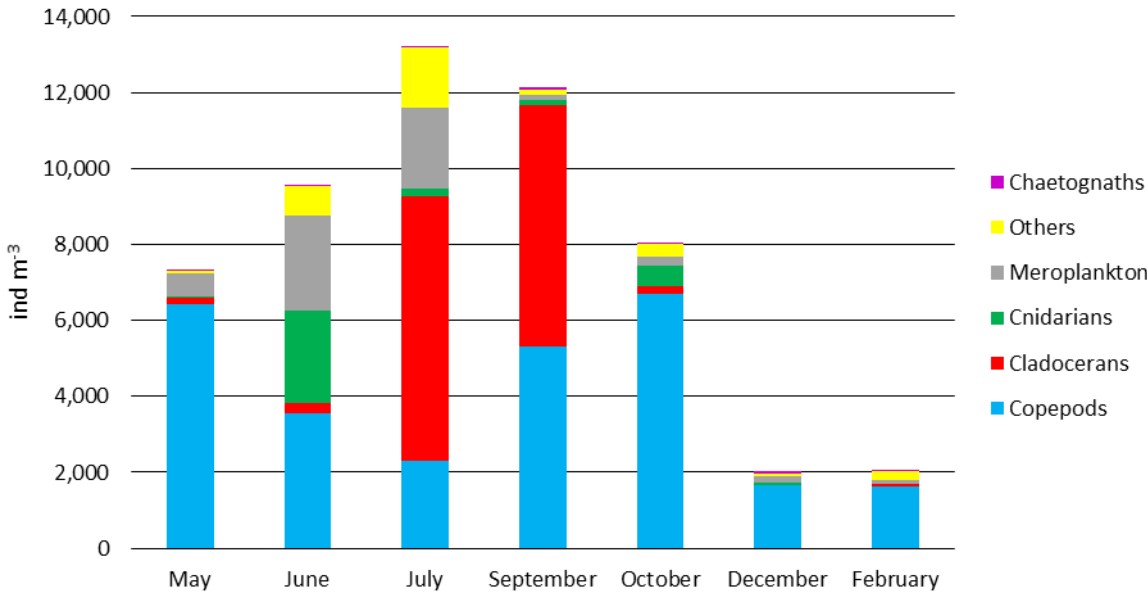

**Figure 3.** Composition and abundance (ind. m$^{-3}$) of mesozooplanktonic taxa in the field during the sampling periods.

### 3.2. Diel Feeding Cycle

The diel pattern of stomach fullness confirmed the diurnal feeding behaviour of *S. pilchardus*. The fullness index (F) was generally low in the morning, gradually increased at noon and peaked at dusk. Overnight, feeding activity had nearly ceased by sunrise. In all months studied, the highest values of stomach fullness were observed between 18:00 and 21:00 and the lowest values around 4:00. The mean Fullness index varied from a minimum of 0.04 in June and July to a maximum of 1.9 in May (Figure 4). In May, the maximum fullness value was twice as high as the maximum values observed in June and

July. In October, it was not possible to describe feeding periodicity because catches did not yield sufficient numbers throughout the day. No sardines were caught in December and February.

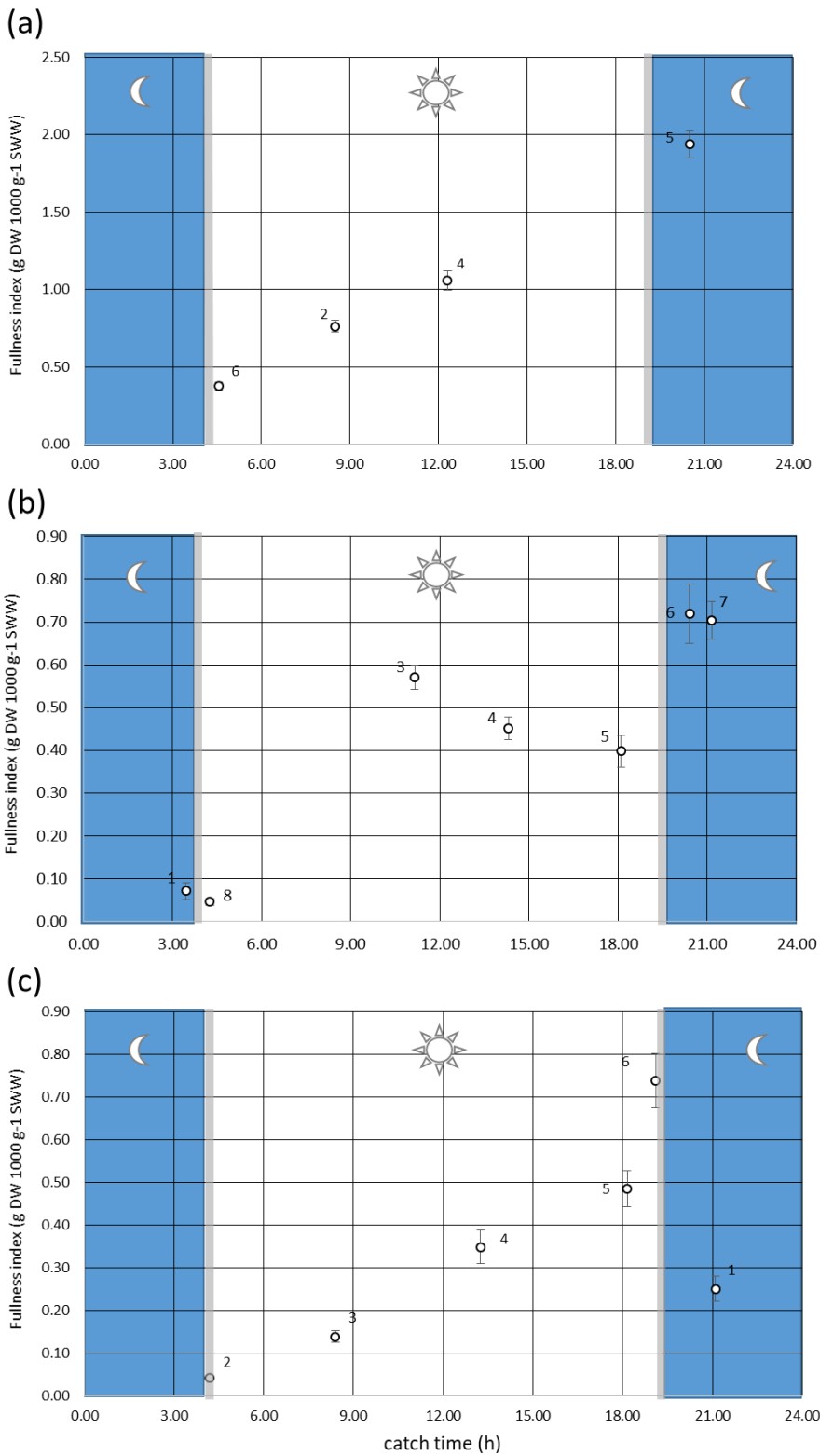

**Figure 4.** Sampling time vs. Fullness index, where numbers refer to the tow number, bars to the standard error and grey shaded line to the sunrise and sunset: (**a**) May; (**b**) June; (**c**) July 2006.

### 3.3. Composition of the Diet

A total of 96 sardines were analysed: 24 specimens collected in the same tow in May, June, July and October (Table 2). The total length (TL) of adult *S. pilchardus* ranged from a minimum of 138 mm to a maximum of 200 mm, with a mean of 169.5 ± 12.4 mm (Table 2).

A total of 15,109 prey items belonging to 73 taxa were identified (Table 3). The number of prey varied from a minimum of 305 to a maximum of 3318 prey/stomach, with an overall mean of 1259 ± 884 prey/stomach. Large differences were observed between months (Table 4). The numerically most abundant prey categories (N%) were copepods (85.0%) and Dinophyceae (9.5%) in May; tintinnids (56.4%), eggs and larvae of teleosts (12.2%), Dinophyceae (11.3%) and copepods (10.8%) in June; copepods (48.4%) and Crustacea larvae (21.1%) in July; copepods (50.4%) and chaetognaths (44.3%) in October.

**Table 3.** Taxonomic composition of identifiable stomach contents expressed as mean ± st. dev. of prey number/stomach.

| Group | Prey Item | 10 May 2006 | 20 June 2006 | 26 July 2006 | 26 October 2006 |
|---|---|---|---|---|---|
| Bacillariophyceae | *Coscinodiscus* spp. | 5.3 ± 2.3 | 2.7 ± 3.1 | 7.0 ± 3.6 | 1.0 ± 1.0 |
| | *Pleurosigma* spp. | 0.0 | 0.0 | 2.0 ± 3.5 | 0.3 ± 0.6 |
| | *Thalassiosira* spp. | 0.0 | 0.7 ± 1.2 | 3.3 ± 5.8 | 0.0 |
| Dinophyceae | *Ceratium candelabrum* | 0.0 | 14.3 ± 10.2 | 0.0 | 0.0 |
| | *Ceratium furca* | 0.0 | 0.0 | 0.0 | 0.3 ± 0.6 |
| | *Ceratium trichoceros* | 0.0 | 12.7 ± 18.6 | 0.0 | 0.0 |
| | *Ceratium tripos* | 1.3 ± 2.3 | 5.3 ± 2.1 | 0.0 | 0.0 |
| | *Ceratium* spp. | 1.3 ± 2.3 | 1.7 ± 2.9 | 0.0 | 0.0 |
| | *Diplopsalis* spp. | 16.0 ± 17.4 | 3.3 ± 3.1 | 2.0 ± 3.5 | 0.0 |
| | *Dinophysis caudata* | 0.0 | 16.0 ± 6.6 | 6.7 ± 9.9 | 0.0 |
| | *Dinophysis fortii* | 0.0 | 0.0 | 0.0 | 0.7 ± 1.2 |
| | *Dinophysis sacculus* | 0.0 | 0.3 ± 0.6 | 0.0 | 0.0 |
| | *Gonyaulax polygramma* | 0.0 | 0.3 ± 0.6 | 0.0 | 0.0 |
| | *Gonyaulax* spp. | 1.3 ± 2.3 | 1.0 ± 1.7 | 2.0 ± 3.5 | 0.0 |
| | *Lingulodinium polyedrum* | 1.3 ± 2.3 | 6.3 ± 3.8 | 9.3 ± 12.9 | 0.3 ± 0.6 |
| | *Podolampas palmipes* | 0.0 | 0.0 | 0.0 | 0.3 ± 0.6 |
| | *Prorocentrum micans* | 85.3 ± 74.4 | 2.7 ± 2.3 | 0.3 ± 0.6 | 0.0 |
| | *Protoperidinium claudicans* | 0.0 | 4.3 ± 2.1 | 0.3 ± 0.6 | 0.0 |
| | *Protoperidinium conicum* | 0.0 | 12.7 ± 5.9 | 11.0 ± 6.1 | 0.0 |
| | *Protoperidinium crassipes* | 109.3 ± 26.6 | 54.3 ± 39.5 | 23.0 ± 5.6 | 0.7 ± 1.2 |
| | *Protoperidinium depressum* | 6.7 ± 4.6 | 2.0 ± 2.0 | 2.3 ± 3.2 | 2.0 ± 2.6 |
| | *Protoperidinium divergens* | 8.0 ± 8.0 | 15.0 ± 8.5 | 0.0 | 0.0 |
| | *Protoperidinium oblongum* | 0.0 | 3.3 ± 1.2 | 0.0 | 0.0 |
| | *Protoperidinium oceanicum* | 2.7 ± 2.3 | 0.3 ± 0.6 | 0.3 ± 0.6 | 0.0 |
| | *Protoperidinium steinii* | 0.0 | 1.0 ± 0.0 | 2.0 ± 3.5 | 0.0 |
| | *Ornithocercus magnificus* | 0.0 | 0.3 ± 0.6 | 0.0 | 0.0 |
| Tintinnina | *Codonellopsis schabi* | 0.0 | 0.0 | 0.0 | 6.0 ± 7.8 |
| | *Eutintinnus fraknoii* | 0.0 | 1.0 ± 1.0 | 0.0 | 0.0 |
| | *Stenosemella ventricosa* | 0.0 | 16.0 ± 9.0 | 0.0 | 0.0 |
| | *Tintinnopsis radix* | 0.0 | 766.0 ± 313.6 | 0.0 | 0.0 |
| Mollusca | Gastropoda pediveliger | 1.3 ± 2.3 | 5.3 ± 3.8 | 0.7 ± 0.6 | 0.0 |
| | Bivalvia veliger | 1.3 ± 2.3 | 1.3 ± 0.6 | 12.7 ± 10.1 | 1.7 ± 1.5 |
| Annelida | Polychaeta larvae | 0.3 ± 0.6 | 1.0 ± 0.0 | 0.3 ± 0.6 | 0.0 |
| Cladocera | *Evadne nordmanni* | 21.3 ± 23.1 | 9.3 ± 8.7 | 3.3 ± 3.2 | 0.0 |
| | *Evadne spinifera* | 5.3 ± 2.3 | 2.0 ± 0.0 | 0.0 | 0.0 |
| | *Penilia avirostris* | 0.0 | 2.0 ± 2.6 | 5.0 ± 4.4 | 0.0 |
| | *Pleopis polyphemoides* | 2.7 ± 4.6 | 7.0 ± 3.6 | 17.7 ± 7.2 | 0.0 |
| | *Podon intermedius* | 0.0 | 1.0 ± 1.0 | 5.0 ± 4.6 | 0.0 |
| | Podonidae indet. | 1.3 ± 2.3 | 0.0 | 0.0 | 0.3 ± 0.6 |
| Calanoida | *Acartia (Acartiura) clausi* | 2.7 ± 4.6 | 2.3 ± 1.5 | 1.7 ± 0.6 | 0.7 ± 0.6 |
| | *Acartia (Acanthacartia) tonsa* | 0.0 | 0.3 ± 0.6 | 0.0 | 0.0 |
| | *Acartia* spp. | 1.3 ± 2.3 | 0.0 | 0.0 | 0.0 |
| | *Calanus* spp. | 0.0 | 0.3 ± 0.6 | 0.0 | 0.0 |
| | *Centropages ponticus* | 105.3 ± 97.4 | 1.0 ± 1.7 | 0.0 | 0.0 |
| | *Centropages typicus* | 504.0 ± 176.9 | 0.0 | 0.0 | 0.0 |
| | *Centropages* spp. | 13.3 ± 12.2 | 0.0 | 0.0 | 0.0 |
| | *Clausocalanus* sp. | 0.0 | 0.0 | 0.0 | 0.3 ± 0.6 |

**Table 3.** *Cont.*

| Group | Prey Item | 10 May 2006 | 20 June 2006 | 26 July 2006 | 26 October 2006 |
|---|---|---|---|---|---|
| | *Nannocalanus minor* | 0.0 | 0.0 | 0.0 | 0.3 ± 0.6 |
| | *Paracalanus parvus* | 0.0 | 0.3 ± 0.6 | 0.0 | 0.0 |
| | *Paracalanus* spp. | 61.3 ± 56.8 | 0.3 ± 0.6 | 0.0 | 5.3 ± 7.6 |
| | *Temora longicornis* | 25.3 ± 6.1 | 0.7 ± 1.2 | 0.0 | 0.0 |
| | *Temora stylifera* | 21.3 ± 30.3 | 0.7 ± 0.6 | 5.3 ± 2.1 | 4.7 ± 3.2 |
| | Clauso-Paracalanidae | 677.3 ± 394.9 | 65.3 ± 59.8 | 84.7 ± 35.9 | 0.3 ± 0.6 |
| | Calanoida indet. | 546.7 ± 556.7 | 0.0 | 0.0 | 9.7 ± 2.5 |
| Cyclopoida | *Oithona nana* | 0.0 | 19.3 ± 18.8 | 13.7 ± 8.6 | 1.0 ± 1.0 |
| | *Oithona* cf. *nana* | 5.3 ± 6.1 | 0.0 | 0.0 | 1.0 ± 1.7 |
| | *Oithona plumifera* | 1.3 ± 2.3 | 0.7 ± 1.2 | 0.0 | 0.7 ± 1.2 |
| | *Oithona* cf. *plumifera* | 0.0 | 0.0 | 0.0 | 0.7 ± 1.2 |
| | *Oithona setigera* | 0.0 | 0,0 | 0.0 | 1.3 ± 2.3 |
| | *Oithona* cf. *similis* | 2.7 ± 4.6 | 0.0 | 0.0 | 0.0 |
| | *Oithona* spp. | 1.3 ± 2.3 | 0.0 | 0.0 | 2.3 ± 2.5 |
| Ergasilida | Corycaeidae indet. | 58.7 ± 22.7 | 4.3 ± 2.3 | 35.7 ± 6.4 | 33.7 ± 6.7 |
| | Oncaeidae indet. | 14.7 ± 6.1 | 37.7 ± 11.8 | 195.7 ± 11.0 | 135.7 ± 71.4 |
| Harpacticoida | *Clytemnestra scutellata* | 0.0 | 0.0 | 1.0 ± 1.0 | 0.0 |
| | *Euterpina acutifrons* | 18.7 ± 4.6 | 1.3 ± 1.5 | 13.3 ± 5.5 | 25.7 ± 10.0 |
| | *Microsetella rosea* | 8.0 ± 4.0 | 3.0 ± 1.0 | 2.3 ± 1.5 | 6.0 ± 3.6 |
| | Harpacticoida indet. | 2.7 ± 2.3 | 1.0 ± 1.0 | 0.0 | 1.3 ± 1.5 |
| | Copepoda nauplii | 8.0 ± 4.0 | 11.0 ± 4.6 | 2.3 ± 3.2 | 3.3 ± 4.9 |
| Cirripedia | Cirripedia nauplii | 5.3 ± 6.1 | 1.7 ± 0.6 | 53.7 ± 17.6 | 0.0 |
| | Cirripedia cypris | 1.3 ± 2.3 | 5.3 ± 0.6 | 11.7 ± 8.3 | 0.0 |
| Stomatopoda | *Squilla mantis* alima | 0.0 | 0.0 | 0.3 ± 0.6 | 0.0 |
| Amphipoda | Hyperiidae indet. | 0.0 | 0.0 | 0.3 ± 0.6 | 0.0 |
| Decapoda | *Porcellana* zoeae | 0.0 | 0.3 ± 0.6 | 4.7 ± 4.2 | 0.0 |
| | Decapoda nauplii | 1.3 ± 2.3 | 3.0 ± 5.2 | 0.0 | 0.0 |
| | Decapoda zoeae | 1.3 ± 2.3 | 16.7 ± 22.0 | 24.0 ± 10.4 | 0.0 |
| | Decapoda mysis | 1.3 ± 2.3 | 55.8 ± 15.6 | 60.7 ± 28.6 | 3.5 ± 2.5 |
| | Decapoda phyllosoma | 0.0 | 0.0 | 0.3 ± 0.6 | 0.0 |
| Appendicularia | *Oikopleura* spp. | 0.0 | 1.0 ± 1.7 | 9.0 ± 9.0 | 2.0 ± 2.0 |
| Chaetognatha | Chaetognatha indet. | 0.7 ± 0.9 | 5.2 ± 4.4 | 36.2 ± 17.2 | 205.5 ± 101.7 |
| Osteichthyes | *Engraulis encrasicolus* eggs | 1.3 ± 2.3 | 27.3 ± 4.9 | 3.7 ± 2.1 | 0.3 ± 0.6 |
| | Teleostea spheric eggs | 13.3 ± 12.2 | 142.0 ± 11.0 | 2.0 ± 1.0 | 2.3 ± 2.5 |
| | Teleostea larvae | 0.0 | 0.0 | 0.3 ± 0.6 | 0.0 |
| Invertebrata eggs | Invertebrata eggs spheric *a* | 0.0 | 0.3 ± 0.6 | 0.0 | 0.0 |
| | Invertebrata eggs spheric *b* | 61.3 ± 85.4 | 2.0 ± 3.5 | 2.7 ± 3.8 | 4.3 ± 6.7 |
| | Invertebrata eggs elliptical | 1.3 ± 2.3 | 0.0 | 0.0 | 0.0 |
| | Crustacea eggs elliptical | 0.0 | 6.3 ± 5.7 | 18.3 ± 31.8 | 0.0 |
| | Chaetognatha eggs | 5.3 ± 9.2 | 0.0 | 35.0 ± 47.9 | 0.0 |
| | TOTAL | 2446.4 ± 771.7 | 1389.4 ± 382.8 | 734.9 ± 109.4 | 465.7 ± 228.3 |

Copepods were the most abundant food category in the stomachs of captured sardines in all months, with the sole exception of June when tintinnids predominated numerically. Among copepods, the Clauso-Paracalanidae group and the genus *Oncaea* were by far the most important, alternately dominating. Copepod nauplii were always found, with relatively small amounts, from 0.38 to 7.35% of copepods. A variable amount of invertebrate eggs with diameters ranging from 33 to 88 μm, probably belonging to copepods, was observed in the gut contents. Nevertheless, we did not consider these eggs as "prey" because it was impossible to determine whether they were ingested intentionally or as egg masses carried by the captured copepods. Chaetognaths represented an important food category in October in terms of numbers (44.3%), immediately after copepods.

We found microphytoplankton in all seasons with 26 taxa representing numerically about 10% of the prey in May, June and July and only 1% in October. In contrast, tintinnids were present with four taxa and were especially present in June when they dominated the number of prey (55.13%), probably related to a bloom of *Tintinnopsis radix*.

**Table 4.** Composition of *Sardina pilchardus* diet expressed as prey percentage on numerical (N%) and carbon basis, in May, June, July and October 2006.

| | May | | June | | July | | October | |
|---|---|---|---|---|---|---|---|---|
| Prey group | N% | Carbon Content | N% | Carbon Content | N% | Carbon Content | N% | Carbon Content |
| Bacillariophyceae | 0.22 | 0.00 | 0.24 | 0.00 | 1.68 | 0.00 | 0.29 | 0.00 |
| Dinophyceae | 9.54 | 0.04 | 11.32 | 0.06 | 8.07 | 0.02 | 0.93 | 0.00 |
| Tintinnina | 0.00 | 0.00 | 56.36 | 0.29 | 0.00 | 0.00 | 1.29 | 0.00 |
| Mollusca larvae | 0.11 | 0.05 | 0.48 | 0.23 | 1.81 | 0.46 | 0.36 | 0.02 |
| Polychaeta larvae | 0.01 | 0.01 | 0.07 | 0.06 | 0.05 | 0.02 | 0.00 | 0.00 |
| Branchiopoda | 1.25 | 0.24 | 1.54 | 0.34 | 4.22 | 0.65 | 0.07 | 0.00 |
| Calanoida | 80.06 | 94.73 | 5.13 | 3.97 | 12.47 | 7.68 | 4.58 | 0.92 |
| Cyclopoida | 0.44 | 0.06 | 1.44 | 0.09 | 1.86 | 0.06 | 1.50 | 0.07 |
| Ergasilida | 3.00 | 0.84 | 3.02 | 0.86 | 31.48 | 5.23 | 36.36 | 1.19 |
| Harpacticoida | 1.53 | 0.18 | 1.18 | 0.06 | 2.59 | 0.22 | 7.80 | 0.08 |
| Cirripedia larvae | 0.27 | 0.05 | 0.50 | 0.37 | 8.89 | 2.44 | 0.00 | 0.00 |
| Amphipoda | 0.00 | 0.00 | 0.00 | 0.00 | 0.05 | 0.12 | 0.00 | 0.00 |
| Crustacea larvae | 0.16 | 0.46 | 5.46 | 24.03 | 12.25 | 32.76 | 0.75 | 0.38 |
| Appendicularia | 0.00 | 0.00 | 0.07 | 0.02 | 1.22 | 0.11 | 0.43 | 0.01 |
| Chaetognatha | 0.03 | 0.51 | 0.37 | 6.66 | 4.93 | 47.33 | 44.13 | 97.01 |
| Teleostea eggs and larvae | 0.60 | 2.75 | 12.19 | 62.93 | 0.82 | 2.44 | 0.57 | 0.33 |
| Invertebrata eggs | 2.78 | 0.07 | 0.62 | 0.03 | 7.62 | 0.44 | 0.93 | 0.00 |

In May, we observed a very high number of pollen grains (4616 ± 637.4 cells/stomach), but they were not included in the diet analysis. In spring, pollen may be very abundant in coastal surface waters, but we have not yet found evidence of the ability of sardines to digest it.

### 3.4. Food Carbon

An estimation of prey carbon content showed that *S. pilchardus* obtained almost all carbon from metazoans, especially copepods, chaetognaths, crustacean larvae, and the eggs and larvae of teleosts (Table 4). This was true even when considerable amounts of microphytoplankton and microzooplankton were present in the stomach contents. For example, in June, tintinnids were the most abundant prey in the stomach contents (56.4%); however, they accounted for only 0.3% of the total carbon ingested. Zooplanktonic prey of large size and high carbon content were the most important energetic food source, even when only a few specimens were ingested, such as chaetognaths in July (4.9% by number, 47.3% by carbon content), teleost eggs and larvae in June (12.2% by number, 62.9% by carbon content) or Crustacea larvae in June (5.5% by number, 24.0% by carbon content).

Differences among months were even more pronounced when dietary carbon content was considered. In fact, only one prey category accounted for most of the total carbon intake in each month: 97% of chaetognaths in October, 96% of copepods in May, 63% of teleost eggs and larvae in June, and again 47% of chaetognaths in July.

The overall size spectrum of prey ranged from 17 μm for the diatom *Thalassiosira* spp. to 18,388 μm for chaetognaths (Figure 5), but in all seasons most prey had sizes from 400 to 1000 μm. Isolated peaks occurred only in May in the 50–100 μm size range (mainly corresponding to Dinophyceae) and in June in the 200–250 μm size range (mainly corresponding to Tintinnids) (Figure 5). Overall, most of the prey carbon was in size classes from 600 to 1000 μm, especially in May and June, with an isolated peak in June and July in the 1700–1750 μm size class (corresponding to Crustacea larvae).

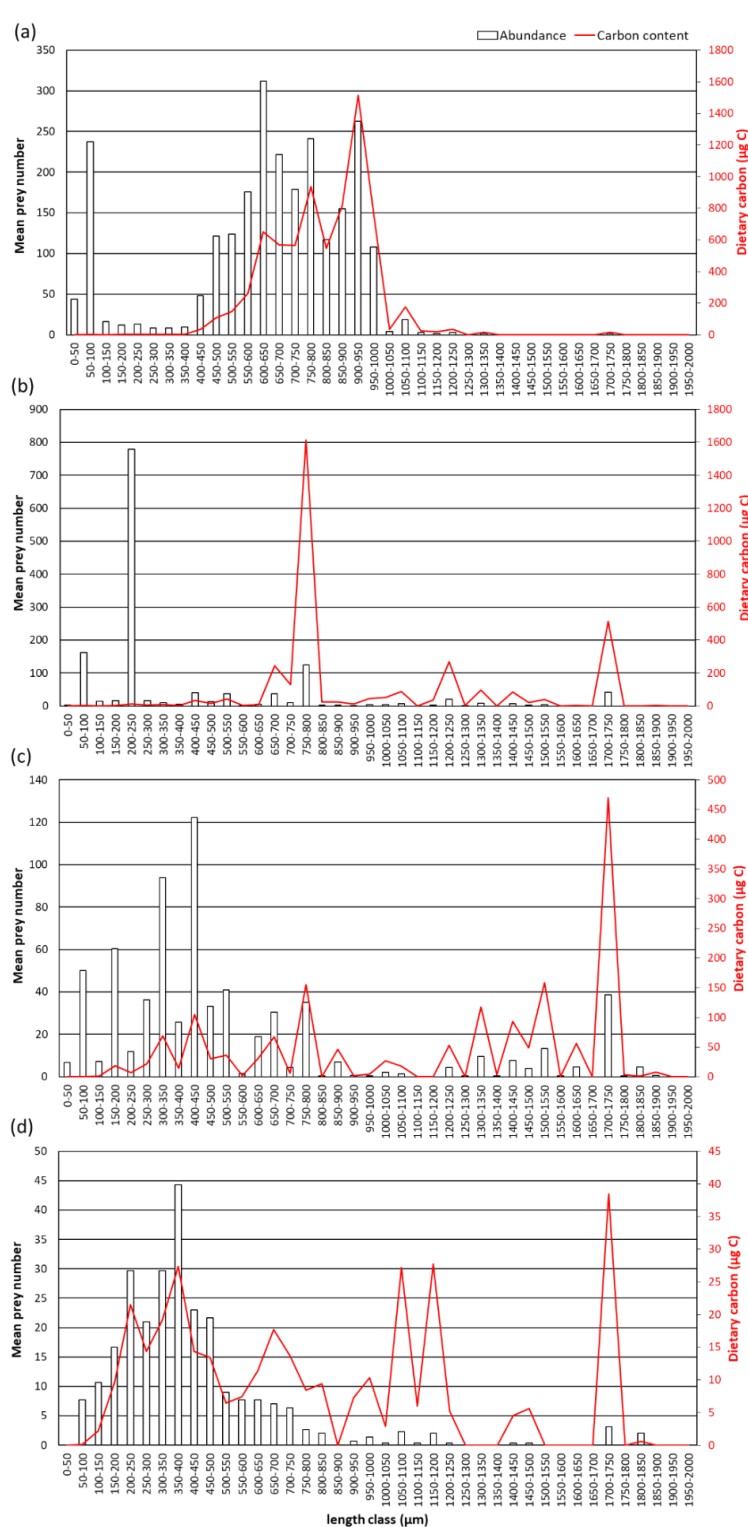

**Figure 5.** Distribution of small prey (<2 mm size) in gut contents: mean counted prey/stomach by size classes of 50 µm (histograms) and their relative carbon content (red solid line) in (**a**) May, (**b**) June, (**c**) July and (**d**) October.

### 3.5. Selection of the Prey

The Ivlev index was calculated for 16 prey groups, excluding those prey (Bacyllario-phyceae, Dinophyceae, Tintinnina, eggs of Invertebrata) that were too small to be effectively retained by the mesh of the WP2 plankton net. All taxa found at abundances <1% in both

food and environment (e.g., ctenophores, nemerteans, phoronids, ostracods, stomatopods, amphipods, isopods, mysidaceans, thaliaceans, cephalochordates) were grouped into a category labelled "Others".

The Ivlev index values (Figure 6) confirmed the preference of adult sardines for large prey such as chaetognaths, decapod larvae, eggs and larvae of teleosts. Nevertheless, some smaller prey such as harpacticoid and cyclopoid (Ergasilida) copepods (with sizes of 150–880 μm and 125–625 μm, respectively) and cirripede larvae (220–660 μm) were also positively selected. Other prey items were selected only occasionally (copepod nauplii, cyclopoid copepods and Cladocera).

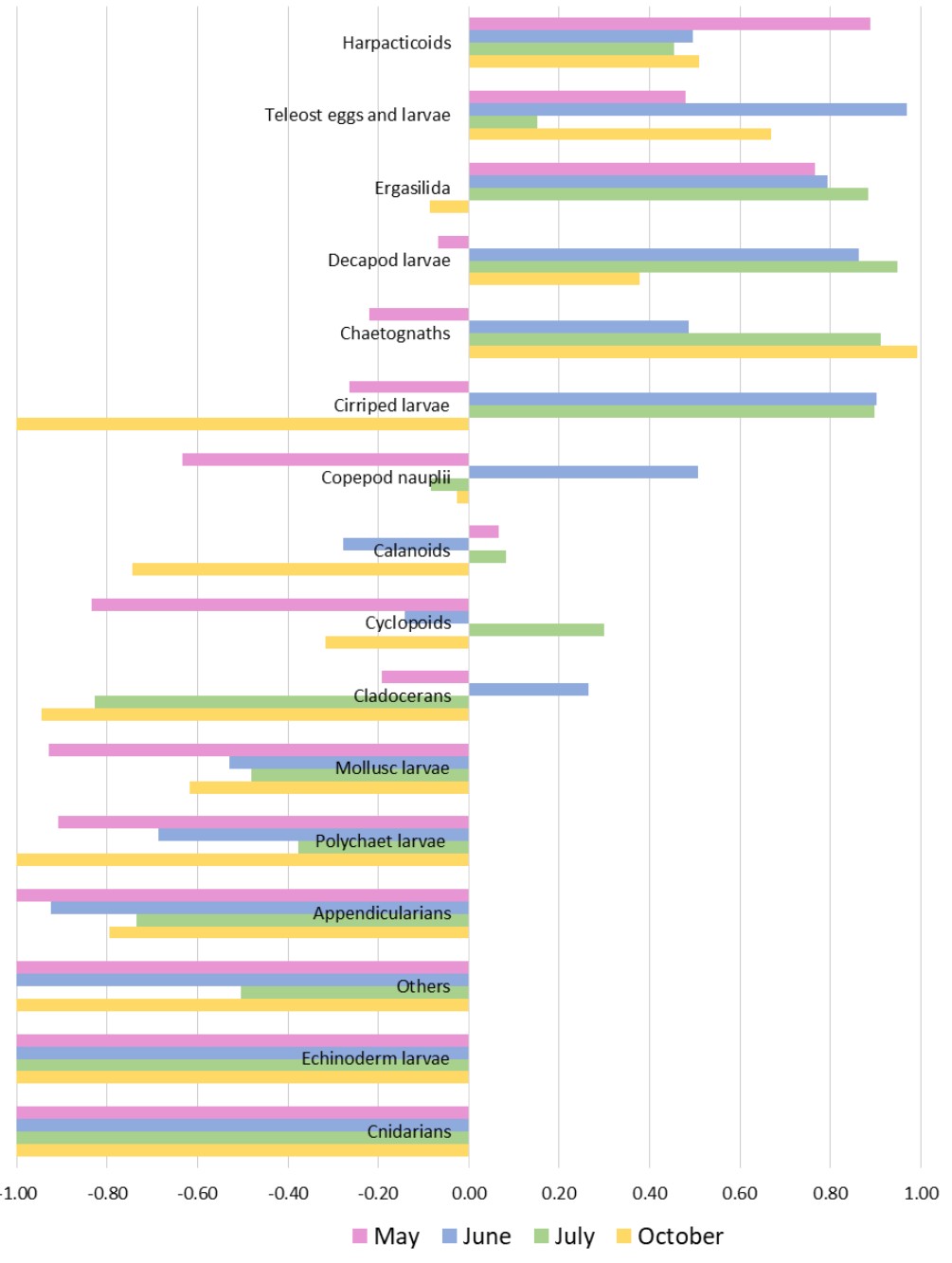

**Figure 6.** Graphical representation of Ivlev's electivity index calculated for prey groups at each sampling period.

Completely avoided potential prey (mesozooplanktonic specimens of taxa found in the sea but never observed in gut contents) that had negative Ivlev index values were cnidarians and echinoderm larvae. Partially avoided prey (mesozooplanktonic specimens of taxa more abundant at sea than in the gut contents, in terms of relative abundance) were: "Others", Appendicularia, mollusk larvae and polychaetes larvae.

## 4. Discussion

### 4.1. Trophic Environment

In the Adriatic Sea, *Sardina pilchardus* spawns from October to May at water temperatures between 9 and 20 °C, with peaks between 11 °C and 16 °C, and salinity between 35.2 and 38.8 [35]. Gamulin and Hure [36] noted that sardines disappear from fishing grounds from October to March, which corresponds to their spawning season. Tičina et al. [37] suggested that sardines leave the most productive shallow waters of the north in the fall for the deeper waters of the south, where they find the stable and relatively warm waters necessary for spawning. This is probably the reason why we did not catch any fish in winter. Vučetić [23] also reported that she had difficulty obtaining adequate samples of sardines in winter.

According to other authors, sardine migration is favoured by the search for the optimal trophic conditions, with zooplankton concentration being higher in May–June in the shallow coastal areas, especially meroplankton [24,38], while in winter it becomes more available in the upper layers of the offshore areas, especially for large copepods [39–42]. In any case, migration to open and deeper waters is assumed to be associated with spawning and overwintering [43]. In the study area, copepods were the most abundant mesozooplanktonic organisms, with the only exception being in summer when cladocerans (with the summer swarming species *Penilia avirostris*) predominated in numbers. Meroplankton, consisting mainly of echinoderms, mollusk and crustacean larvae, were abundant in the spring. This is the typical seasonal outline of the neritic and estuarine plankton community in the northern Adriatic [44].

### 4.2. Diel Feeding Cycle

In May, June and July, sardines exhibited high stomach fullness during the dusk and early night hours. Evidence of daytime feeding by *S. pilchardus* has been previously observed in the Adriatic Sea, both in adults [23] and late larvae [45], as well as in the northern Aegean Sea [16] and in the Catalan Sea [19]. The late afternoon feeding peak coincides with the migration of zooplankton to the surface. Zooplankton species are nearly transparent, so only some specific pigmented body parts can be seen by a visual predator [46]. Such prey are difficult to detect in light with a natural angular distribution (low image contrast). Nevertheless, planktivorous fish may increase the contrast of their prey by searching for it at an angle greater than 48.6° to the vertical, as this makes the prey appear bright against a dark background [47]. This finding could explain the feeding peak that occurs when the sun is generally low on the horizon and the angle to the vertical is greater.

When analysing the stomach contents of fish caught by commercial purse seines operated at night under artificial light, a fundamental problem arises because artificial lights attract different species of mesozooplankton in different ways, creating unnatural conditions. This leads to a bias as the natural diet of small pelagic fishes is described under artificial conditions: both qualitative and quantitative aspects are distorted.

### 4.3. Diet Composition and Seasonality

The decision to analyse only samples caught in the late afternoon, during intense feeding activity, was justified by the possibility to detect prey not yet digested and to describe the composition of the bulk of the diet. We identified an average number of 466 to 2446 prey items per stomach. This abundance is comparable to the results obtained by Nikolioudakis et al. [17] in the northern Aegean (from 83 to 3334) and by Costalago

et al. [15] in the western Mediterranean (from 1498 in winter to 4843 in summer). In contrast, the results are quite different when compared to those of Costalago et al. [15] in the Iberian Atlantic (from 40,126 in winter to 1,231,010 in summer). However, when phytoplankton is excluded, the range found by Costalago et al. [15] in the Atlantic (1339 prey/stomach in winter and 4094 prey/stomach in summer) is more similar to that found in this work in the northern Adriatic (from 460 to 2208 prey/stomach).

The results of the present study show that the diet of adult *S. pilchardus* in the northern Adriatic consists mainly of zooplankton. The diet was numerically based on copepods in all months, with the only exception being in June when the diet was more differentiated. Among the copepods, the Clauso-Paracalanidae group was the most abundant, followed, in decreasing order, by the genus *Centropages*, unidentified Calanoida, the families Oncaeidae and Corycaeidae, the species *Euterpina acutifrons* and the genera *Temora* and *Oithona*. These copepod groups have also been reported by other authors as important prey on the Spanish Atlantic coast [48], on the Portuguese coast [49], in the northwestern Mediterranean [6,15,19], in the northern Aegean [17], in the eastern Aegean [21] and in the central Adriatic [23,26].

Other prey categories reached high abundances only in a single month: tintinnids (in June), decapod larvae (in July) and chaetognaths (in October). Tintinnids have been confirmed as food for sardine in the Mediterranean [50] and on the Atlantic coast of Spain [49,51,52]. Crustacean larvae have also been found by other authors [15,17,19,21,23,26,49]. In contrast, chaetognaths have only been observed by Vučetić [23] in the central Adriatic. Teleostea eggs were also found in sardine stomachs in previous studies [49,51,53], but only in small quantities in the Mediterranean [15,26].

Dinoflagellates (Dinophyceae) were consistently present in the diet from May to July, while diatoms (Bacillariophyceae) were found in surprisingly low amounts. We observed a mean of 113.6 phytoplanktonic cells/stomach, a value lower than those found by Costalago et al. [15] in the northwestern Mediterranean (from 364.8 to 1192.8) and especially on the Iberian Atlantic coast (from 38,787.1 to 1,226,915.8).

### 4.4. Dietary Carbon

Early studies on the diet of *Sardinops sagax* from the Benguela region, based on abundance data or the volumetric method, indicated that sardines were phytoplanktivorous, non-selective filter feeders [54]. Later studies have shown that zooplankton make up the largest proportion of the diet, although phytoplankton play an important role in certain regions (upwelling systems) or in particular periods of the year [55,56]. The contribution of phytoplankton to the carbon content of the diet of adult sardines varies widely, ranging from 14–19% on Portuguese coasts [49] to <3% in the northern Aegean [17]. In the northern Adriatic, we found that the carbon uptake of sardine completely relies on zooplankton, while the contribution of phytoplankton to the total carbon content of the diet was 0.02%. The main food categories in terms of carbon content were: chaetognaths (49.9%), copepods (30.8%), teleost eggs and larvae (10.3%) and decapod larvae (8.0%). It should be noted that in the present study, the contribution of copepods to the carbon content of the diet was underestimated as their egg masses were not taken into account.

The seasonality of the diet, expressed as carbon content, showed even more marked differences than the number of prey. Copepods, chaetognaths, eggs and larvae of teleosts and decapod larvae accounted for almost all the carbon content of the diet in all seasons considered. Similarly, Costalago and Palomera [19] found that, regardless of their numerical importance, decapod larvae and copepods contributed more than any other prey to seasonal differences in diet.

### 4.5. Feeding Selectivity

Although filter feeding is considered the main feeding mode in *S. pilchardus* [48,57], the ability to switch to a particulate feeding mode was described by Garrido et al. [49] for sardines under experimental conditions: filtering was adopted when small particles ($\leq$724 μm)

were offered as food, while particulate feeding started when bigger prey ($\geq$780 μm) were proposed. When a wide range of prey sizes was offered, both feeding modes were simultaneously adopted. In the studied area, prey showed a wide range of sizes (from 17 μm to 18,388 μm) without a clear distribution (Figure 6), suggesting that adult sardines in the northern Adriatic may use both filter and particulate feeding regardless of the season.

The Ivlev index pointed out that sardines actively selected teleost eggs and larvae, decapod larvae and chaetognaths, prey that were less abundant in the plankton samples than in stomach contents. Nevertheless, not only large prey were positively selected, but also small copepods such as *Oncaea* spp. and *Euterpina acutifrons* (whose maximum size is <750 μm) were preferred. The high selectivity for these small copepods reported in this and other works [17,19] and even for anchovies [58,59], could probably be explained by their tendency to associate with detritus and/or gelatinous zooplankton (e.g., [60,61]), which induce them to aggregate into patches, making them easy prey to be pursued and caught by filtering sardines.

Further evidence of selectivity in food intake is the fact that cladocerans were only marginally present in the stomach contents, even though they were dominant in the summer plankton. This result contrasts with Costalago and Palomera [19], who found that cladocerans are highly selected by sardines in the northwestern Mediterranean during summer. The importance of cladocerans in the diet of small pelagic fishes is not clear: some authors emphasise their importance [18,62,63], but others found that high concentrations of cladocerans could even have unfavourable effects on the feeding activity of small planktivores [64].

Some of the most abundant meroplankton (echinoderm larvae) were never found in the stomachs and thus were completely avoided by the sardines. The inconsistency of prey composition in stomach contents in relation to plankton has also been noted by other authors [17,19,23]. In laboratory studies, sardines, when fed with wild mixed prey assemblages, also showed a preference for copepods and decapods over other zooplankton [11]. The constant presence of copepods in gut contents suggests that certain prey characteristics may be more likely to induce predation by fish. In planktivores, prey detection may be strongly influenced by prey movement [65,66], shape and colour of prey body [67], relative orientation between prey and predator [68] and light intensity [69].

The observed low selectivity for calanoid copepods could be due to the ability of these prey to escape sardine predation [17,70]. The swimming behaviour of copepods generally varies between continuous and intermittent locomotion, but can also have even more complex features, as in the genus *Clausocalanus*, involving a rapid and continuous movement in intertwined small loops [71]. The short pauses of motion may provide copepods with brief moments of invisibility to predators, and the subsequent sinking may increase their perceptual ability [72,73]. In addition, copepod species appear to modify their escape behaviour depending on the strength of the stimulus they encounter [74].

Explaining food selection in small planktivorous fishes is quite difficult as little is known so far about the vertical distribution of planktonic species, their ability to swarm and their escape capacity. This, in turn, complicates considerations about the probability of an encounter between prey and predator. The predator is often able to compensate for the prey's adaptations, resulting in equal capture success [74]. In any case, for sardines, the factor determining the selection/avoidance of a potential prey, as well as the switch from filter feeding to the particulate feeding mode, is the final energy uptake achieved when the gains from consumption exceed the losses from capture. On the other hand, sardines can influence planktonic community structures and food web functioning thanks to their ability to select for food.

Studying the adaptive capacity of small pelagic fishes is central to better management of fisheries' resources. Given the importance of mesozooplankton in understanding the ecology of small pelagic fishes, scientific surveys should be carried out with regular environmental monitoring, including plankton sampling.

**Author Contributions:** Conceptualization, D.B. and V.T.; methodology, D.B., A.d.O. and S.L.; investigation, D.B., S.L. and V.T.; writing—original draft preparation, D.B. and V.T.; writing—review and editing, D.B., A.d.O., S.L. and V.T.; supervision, V.T. All authors have read and agreed to the published version of the manuscript.

**Funding:** This research was funded by Project EcoMAdr (INTERREG IIIA Italy–Slovenia) and the Flagship Project RITMARE—The Italian Research for the Sea—coordinated by the Italian National Research Council and funded by the Italian Ministry of Education, Universities, and Research within the National Research Program 2011–2013. D.B. and S.L. were funded by the Project EcoMAdr (INTERREG IIIA Italy–Slovenia).

**Institutional Review Board Statement:** Not applicable.

**Informed Consent Statement:** Not applicable.

**Data Availability Statement:** All data generated during this study are included in this article with the exception of temperature and salinity data which are available on request from the corresponding author.

**Acknowledgments:** Special thanks to the crew of Cuba and Acquario fishing vessels for their assistance in the field work and to Alenka Goruppi for the revision of the taxonomical list of zooplankton and the help for the editing of figures. We thank the three anonymous reviewers of this manuscript for their useful suggestions.

**Conflicts of Interest:** The authors declare no conflict of interest. The funders had no role in the design of the study; in the collection, analyses, or interpretation of data; in the writing of the manuscript, or in the decision to publish the results.

## Appendix A

**Table A1.** Morphometric relationships used to calculate (a) dry mass (DM) (µg) and (b) carbon content (µg C) of *Sardina pilchardus* preys. PL: prosome length (µm); L: total length (µm); V: volume. Brackets (a) and (b) indicate which sources the relationships refer to.

| Prey item | (a) | Dry mass (µg) | (b) | Carbon content (µg) | References |
|---|---|---|---|---|---|
| *Cocconeis* spp.<br>*Coscinodiscus* spp.<br>*Diploneis* spp.<br>*Paralia sulcata*<br>*Pleurosigma* spp.<br>*Nitzschia* spp.<br>*Thalassiosira* spp.<br>Pennate diatom | | volume | | C = V × 0.11 | (a) [75,76]; (b) [77] |

**Table A1.** *Cont.*

| Prey item | (a) | Dry mass (µg) | (b) | Carbon content (µg) | References |
|---|---|---|---|---|---|
| *Ceratium candelabrum*<br>*Ceratium furca*<br>*Ceratium trichoceros*<br>*Ceratium tripos*<br>*Ceratium* spp.<br>*Diplopsalis* spp.<br>*Dinophysis caudata*<br>*Dinophysis fortii*<br>*Dinophysis sacculus*<br>*Gonyaulax polygramma*<br>*Gonyaulax* spp.<br>*Katodinium* spp.<br>*Lingulodinium polyedrum*<br>*Phalacroma* spp.<br>*Podolampas palmipes*<br>*Prorocentrum micans*<br>*Protoperidinium claudicans*<br>*Protoperidinium conicum*<br>*Protoperidinium crassipes*<br>*Protoperidinium depressum*<br>*Protoperidinium divergens*<br>*Protoperidinium oblongum*<br>*Protoperidinium oceanicum*<br>*Protoperidinium steinii*<br>*Ornithocercus magnificus*<br>*Scrippsiella* spp.<br>*Protoperidinium* spp. | | volume | | $C = V \times 0.13$ | (a) [75,76]; (b) [77] |
| *Codonellopsis schabi*<br>*Eutintinnus fraknoii*<br>*Stenosemella ventricosa*<br>*Tintinnopsis radix* | | volume | | $C = (444.5 + 0.053 \times V) \times 10^{-6}$ | (b) [78] |
| Gastropoda pediveliger | | direct weight | | $C = (DW \times 31.25)/100$ | (b) [55] |
| Bivalvia veliger | | direct weight | | $C = (DW \times 31.25)/100$ | (a) [79]; (b) [55] |
| Polychaeta larvae | | direct weight | | $C = (DW \times 40)/100$ | (a) [79]; (b) [80] |
| *Evadne nordmanni*<br>*Evadne spinifera*<br>*Penilia avirostris*<br>*Pleopis polyphemoides*<br>*Podon intermedius*<br>Podonidae | | direct weight | | $C = (DW \times 33.1)/100$ | (a) [81]; (b) [82] |
| *Acartia clausi*<br>*Acartia tonsa*<br>*Acartia* spp. | | | | $\text{Log } C = 3.032 \text{ Log PL} - 8.556$ | (b) [83] |
| *Calanus* spp.<br>(ref. *Calanus helgolandicus*) | | $\text{Log DW} = 2.691 \text{ Log PL} - 6.883$ | | $C = 0.372 \text{ DW} - 0.248$ | (a) [84]; (b) [85] |
| *Centropages ponticus*<br>*Centropages typicus*<br>*Centropages* spp. | | $\text{Log DW} = 2.451 \text{ Log PL} - 6.103$ | | $C = (DW \times 37.6)/100$ | (a) [84]; (b) [82] |
| *Clausocalanus* sp.<br>*Paracalanus parvus*<br>*Paracalanus* spp.<br>Clauso-Paracalanidae<br>Calanoida indet. | | | | $\text{Log } C = 3.128 \text{ Log PL} - 8.451$ | (b) [86] |

**Table A1.** *Cont.*

| Prey item | (a) | Dry mass (µg) | (b) | Carbon content (µg) | References |
|---|---|---|---|---|---|
| *Nannocalanus minor* (rif. *Calanus helgolandicus*) | | Log DW = 2.691 Log PL − 6.883 | | C = 0.372 DW − 0.248 | (a) [84]; (b) [85] |
| *Temora longicornis* | | Log DW = 3.059 Log PL − 7.682 | | C = (DW × 46.8)/100 | (a) [84]; (b) [87] |
| *Temora stylifera* | | (Log DW = 2.71 Log L − 3.685)/1000 | | C = (DW × 46.8)/100 | (a) [88]; (b) [87] |
| *Oithona nana* *Oithona* cf. *nana* *Oithona plumifera* *Oithona* cf. *plumifera* *Oithona setigera* *Oithona* cf. *similis* *Oithona* spp. | | | | C = 9.4676 $10^{-7}$ $PL^{2.16}$ | (b) [89] |
| *Corycaeus* spp. (ref. Cyclopoida) | | Ln DW = 1.96 Ln PL − 11.64 | | C = (DW × 43.1)/100 | (a) [90]; (b) [87] |
| *Oncaea* spp. | | direct weight | | C = (DW × 38.2)/100 | (a) (b) [87] |
| *Clytemnestra scutellata* | | Ln DW = 1.96 Ln PL − 11.64 | | C = (DW × 42.4)/100 | (a) [90]; (b) [91] |
| *Euterpina acutifrons* | | DW = 1.389 $10^{-8}$ $L^{2.857}$ | | C = (DW × 46)/100 | (a) (b) [92] |
| *Microsetella rosea* Harpacticoida indet. (ref. *Microsetella norvegica*) | | | | C = 2.65 $10^{-6}$ $L^{1.95}$ | (b) [93] |
| Copepoda eggs | | $4/3\,\pi\,(L/2)^3$ | | C = 140 × $10^{-9}$ × V | (b) [94] |
| Copepod nauplii (ref. *Acartia* nauplii) | | Log DW = 2.848 Log L − 7.265 | | C = (DW × 42.4)/100 | (a) [95]; (b) [91] |
| Cirripedia nauplii | | DW = 80.627 × $L^{4.27}$ | | C = (DW × 39.97)/100 | (a) [55]; (b) [96] |
| Cirripedia cypris | | direct weight | | C = (DW × 39.97)/100 | (b) [96] |
| *Squilla mantis* alima (ref. *Lucifer reynaudii*) | | direct weight | | C = (DW × 41.1)/100 | (a) (b) [97] |
| Hyperiidae indet. *Porcellana* zoeae Decapoda zoeae Decapoda mysis Decapoda phyllosoma | | direct weight | | C = (DW × 43.69)/100 | (a) [79]; (b) [87] |
| Decapoda nauplii (ref. Cirripedia larvae) | | direct weight | | | (a) [98] |
| *Oikopleura* spp. | | | | C = 0.04 × $L^{3.29}$ | (b) [99] |
| Chaetognatha (ref. *Sagitta elegans*) | | DW = 0.114 × $L^{3.1963}$ | | C = (DW × 35.8)/100 | (a) (b) [100] |
| *Engraulis encrasicolus* eggs Teleostea spheric eggs (ref. *Engraulis mordax*) | | direct weight | | C = 0.457 × DW | (a) [101]; (b) [102] |
| Teleostea larvae (ref. *E. encrasicolus* larvae) | | DW = 6 × $10^{-15}$ $L^{4.1229}$ | | C = 0.457 × DW | (a) [58]; (b) [102] |
| Invertebrata spheric eggs (ref. Copepoda eggs) | | $4/3\,\pi\,(L/2)^3$ | | C = 140 × $10^{-9}$ × V | (b) [94] |
| Invertebrata elliptical eggs (ref. Copepoda eggs) | | $4/3\,\pi\,(L/2) \times (L/4)^2$ | | C = 140 × $10^{-9}$ × V | (b) [94] |

**Table A2.** Mesozooplankton abundance (ind m$^{-3}$) and species composition during the studied period: from May 2006 to February 2007. Zooplankton samples were collected at the same time as sardines. Date (day/month/year), time (24 h, GMT + 1).

| | Date | 10 May 06 | 20 June 06 | 26 July 06 | 04 Sept 06 | 26 Oct 06 | 14 Dec 06 | 01 Feb 07 |
|---|---|---|---|---|---|---|---|---|
| | Time | 22:30 | 21:30 | 18:50 | 16:50 | 17:50 | 15:20 | 18:50 |
| Group | Sampling depth (m) | 16 | 16 | 16 | 17 | 20 | 12 | 22 |
| Dinophyceae | *Noctiluca scintillans* | 0 | 20 | 286 | 74 | 0 | 30 | 88 |
| Hydrozoa | Anthomedusae indet. | 27 | 27 | 20 | 85 | 0 | 0 | 0 |
| | *Obelia* spp. | 0 | 12 | 4 | 0 | 6 | 0 | 4 |
| | Leptomedusae indet. | 0 | 4 | 0 | 0 | 0 | 0 | 0 |
| Siphonophorae | *Muggiaea* spp. | 0 | 220 | 0 | 0 | 0 | 0 | 0 |
| | Siphonophorae indet. | 14 | 2141 | 192 | 41 | 527 | 37 | 16 |
| | Hydrozoa indet. | 0 | 0 | 0 | 0 | 0 | 1 | 0 |
| Scyphozoa | Scyphozoa ephyrae | 0 | 0 | 0 | 0 | 0 | 1 | 0 |
| Anthozoa | *Cerianthus* larvae | 0 | 0 | 0 | 0 | 0 | 0 | 1 |
| Ctenophora | Ctenophora larvae | 10 | 31 | 0 | 0 | 0 | 0 | 0 |
| | Ctenophora indet. | 0 | 0 | 0 | 0 | 0 | 4 | 1 |
| Nemertea | Nemertea pilidia | 0 | 20 | 0 | 0 | 0 | 0 | 0 |
| Phoronida | Phoronida actinotrochae | 2 | 0 | 0 | 0 | 0 | 0 | 0 |
| Gastropoda | *Creseis clava* | 0 | 0 | 0 | 0 | 3 | 4 | 0 |
| | Gastropoda pediveligers | 239 | 408 | 455 | 0 | 13 | 60 | 17 |
| Bivalvia | Bivalvia veligers | 6 | 67 | 357 | 37 | 110 | 98 | 34 |
| Polychaeta | Polychaeta larvae | 0 | 12 | 16 | 18 | 56 | 25 | 4 |
| | Polychaeta indet. | 24 | 106 | 0 | 0 | 0 | 0 | 0 |
| Branchiopoda | *Pleopis polyphaemoides* | 2 | 24 | 35 | 4 | 0 | 0 | 37 |
| | *Podon intermedius* | 2 | 16 | 20 | 0 | 6 | 0 | 10 |
| | *Evadne nordmanni* | 78 | 8 | 251 | 129 | 0 | 0 | 0 |
| | *Evadne spinifera* | 73 | 192 | 43 | 122 | 9 | 0 | 0 |
| | *Evadne tergestina* | 0 | 12 | 78 | 0 | 0 | 0 | 0 |
| | Podonidae indet. | 0 | 0 | 0 | 0 | 3 | 0 | 0 |
| | *Penilia avirostris* | 0 | 20 | 6533 | 6097 | 198 | 16 | 1 |
| Ostracoda | Ostracoda indet. | 0 | 8 | 0 | 4 | 0 | 0 | 1 |
| Calanoida | *Acartia (Acartiura) clausi* | 345 | 298 | 145 | 89 | 41 | 22 | 11 |
| | *Acartia* juv. | 388 | 200 | 184 | 26 | 116 | 93 | 17 |
| | *Anomalocera* sp. | 2 | 0 | 0 | 0 | 0 | 0 | 0 |
| | *Calanus helgolandicus* | 200 | 0 | 0 | 0 | 0 | 3 | 10 |
| | *Calocalanus* spp. | 2 | 0 | 0 | 4 | 0 | 0 | 0 |
| | *Candacia* juv. | 0 | 0 | 0 | 0 | 0 | 0 | 1 |
| | *Centropages ponticus* | 39 | 4 | 0 | 7 | 0 | 0 | 0 |
| | *Centropages typicus* | 86 | 0 | 0 | 0 | 0 | 0 | 0 |
| | *Centropages* spp. | 839 | 525 | 24 | 81 | 0 | 0 | 0 |
| | *Clausocalanus arcuicornis* | 6 | 0 | 0 | 7 | 0 | 3 | 30 |
| | *Clausocalanus furcatus* | 2 | 0 | 0 | 59 | 72 | 0 | 13 |
| | *Clausocalanus* juv. | 51 | 8 | 8 | 85 | 144 | 3 | 54 |
| | *Ctenocalanus vanus* | 114 | 0 | 0 | 0 | 0 | 4 | 448 |
| | *Diaixis pygmaea* | 0 | 0 | 0 | 0 | 0 | 8 | 110 |
| | *Euchaeta hebes* | 0 | 0 | 0 | 0 | 0 | 0 | 1 |
| | *Mecynocera clausi* | 0 | 0 | 0 | 4 | 0 | 0 | 0 |
| | *Paracalanus denudatus* | 649 | 8 | 0 | 55 | 0 | 0 | 0 |
| | *Paracalanus parvus* s.l. | 2110 | 1173 | 702 | 1358 | 1239 | 150 | 317 |

**Table A2.** *Cont.*

| | Date | 10 May 06 | 20 June 06 | 26 July 06 | 04 Sept 06 | 26 Oct 06 | 14 Dec 06 | 01 Feb 07 |
|---|---|---|---|---|---|---|---|---|
| | *Pseudocalanus elongatus* | 120 | 0 | 0 | 0 | 0 | 0 | 0 |
| | *Temora longicornis* | 171 | 4 | 12 | 0 | 0 | 0 | 3 |
| | *Temora stylifera* | 2 | 0 | 24 | 1517 | 370 | 41 | 17 |
| | Calanoida copepodites | 739 | 545 | 553 | 698 | 618 | 169 | 147 |
| Cyclopoida | *Oithona nana* | 51 | 459 | 75 | 140 | 85 | 34 | 6 |
| | *Oithona plumifera* | 0 | 0 | 4 | 85 | 72 | 33 | 19 |
| | *Oithona setigera* | 0 | 4 | 4 | 0 | 0 | 0 | 0 |
| | *Oithona similis* | 153 | 0 | 4 | 15 | 13 | 3 | 3 |
| | *Oithona* spp. | 200 | 118 | 71 | 114 | 72 | 255 | 33 |
| Ergasilida | Corycaeidae indet. | 27 | 4 | 35 | 103 | 348 | 170 | 77 |
| | *Oncaea* spp. | 6 | 102 | 267 | 572 | 3250 | 531 | 257 |
| | *Sapphirina* spp. | 0 | 0 | 0 | 0 | 0 | 1 | 0 |
| Harpacticoida | *Clytemnestra scutellata* | 0 | 0 | 0 | 0 | 0 | 1 | 0 |
| | *Euterpina acutifrons* | 4 | 4 | 125 | 225 | 191 | 77 | 36 |
| | Harpacticoida indet. | 2 | 35 | 8 | 4 | 0 | 1 | 0 |
| Copepoda | Copepoda nauplii | 122 | 78 | 59 | 78 | 63 | 60 | 26 |
| Cirripedia | Cirripedia nauplii | 39 | 8 | 75 | 0 | 13 | 1 | 1 |
| Isopoda | Epicaridea indet. | 0 | 0 | 0 | 0 | 3 | 0 | 3 |
| Decapoda | *Pisidia* larvae | 0 | 12 | 0 | 0 | 0 | 0 | 0 |
| | Brachiura zoea | 0 | 0 | 0 | 4 | 0 | 0 | 1 |
| | Decapoda mysis | 2 | 98 | 43 | 44 | 25 | 4 | 1 |
| | Decapoda zoea | 14 | 12 | 4 | 18 | 3 | 0 | 4 |
| | Decapoda nauplii | 0 | 0 | 4 | 0 | 0 | 0 | 0 |
| Mysida | Mysida indet. | 2 | 0 | 8 | 0 | 3 | 0 | 1 |
| Cumacea | Cumacea indet. | 0 | 0 | 0 | 0 | 0 | 0 | 1 |
| Chaetognatha | *Sagitta* spp. | 4 | 39 | 35 | 48 | 13 | 78 | 44 |
| Echinodermata | Asteroidea larvae | 0 | 0 | 27 | 0 | 0 | 0 | 0 |
| | *Psammechinus* larvae | 14 | 4 | 8 | 0 | 0 | 0 | 0 |
| | Echinoidea plutei | 135 | 1733 | 1012 | 4 | 6 | 5 | 41 |
| | Holothuroidea auricularia | 108 | 47 | 16 | 0 | 0 | 0 | 0 |
| | Echinodermata plutei | 39 | 4 | 0 | 0 | 0 | 0 | 0 |
| Hemichordata | Hemichordata tornariae | 0 | 0 | 0 | 0 | 0 | 0 | 1 |
| Appendicularia | *Oikopleura* spp. | 39 | 557 | 1259 | 63 | 314 | 16 | 117 |
| Thaliacea | *Doliolum* spp. | 0 | 47 | 35 | 0 | 19 | 4 | 4 |
| Cephalochordata | *Branchiostoma lanceolatum* juv. | 0 | 43 | 0 | 0 | 0 | 0 | 0 |
| Vertebrata | Osteichthyes eggs | 14 | 27 | 35 | 4 | 6 | 0 | 0 |
| | Osteichthyes larvae | 4 | 31 | 59 | 4 | 3 | 0 | 1 |
| | TOTAL | 7320 | 9576 | 13,212 | 12,125 | 8035 | 2044 | 2075 |

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
