# Peer review of "Diet of Adult Sardine Sardina pilchardus in the Gulf of Trieste, Northern Adriatic Sea"

_jmse, doi:10.3390/jmse10081012_

Round 1

Reviewer 1 Report

This is a very well-written and very interesting article concerning the diet of Sardina pilchardus in the northernmost coastal area of the Mediterranean Sea. Studies on the trophic ecology of fishes are really essential as they represent fundamental knowledge to improve the assessment and management of commercially important species and not only. Another issue that is really important and imposed in this article, not often investigated nowadays, is the study of fish diet in relation to food availability resulting findings concerning food selectivity. In general, there is lack of information on the abundance and distribution of food/prey species in the environment leading, in several cases, to the conclusion that fishes are characterized by selective feeding behaviour without any consideration of their feeding preferences with respect to prey availability (Koulouri et al., 2015; Maidanou et al., 2021). The objectives of the study are well defined. The article also describes in a detailed and efficient way methods and results of this study. Some more specific comments are indicated in the text. I recommend acceptance for publication of this article with minor revisions.

Koulouri P., Dounas C., Arvanitidis C., Koutsoubas D., Tselepides A., Eleftheriou A., 2015. A. field experiment on trophic relations within the benthic boundary layer (BBL) over an oligotrophic continental shelf. Estuarine, Coastal and Shelf Science, 164, 392-407. https://doi.org/10.1016/j.ecss.2015.07.029

Maidanou M., Koulouri P., Karachle P.K., Arvanitidis C., Koutsoubas D., Dounas C., 2021. Trophic Diversity of a Fish Community Associated with a Caulerpa prolifera (Forsskål) Meadow in a Shallow Semi-Enclosed Embayment. Journal of Marine Science and Engineering, 9, 165. https://doi.org/10.3390/jmse9020165

Author Response

Please see the 2 attachments: one document in word and one pdf

Reviewer 2 Report

This Ms provides a somewhat traditional and very descriptive approach to diet of Sardina pilchardus. Thus, and even though the information is interesting, the degree of innovation of this investigation is limited.

Overall, the Ms is well organized and minor comments suggestions refer to:

Title: Instead of a general, and non-precise, reference to the area sampled (as "... northernmost coastal area of the Mediterranean Sea." a objective reference (such as "Gulf of Trieste, Northern Adriatic") should have been used.

The Abstract is not informative enough about the sample size and the main methods and main conclusion.

Sardina pilchardus must be included in the list of key-words.

In the Introduction the paragraph from lines 33 - 41 should appear after paragraph from lines 43-68 as the former refers to the species considered in this study and the latter to general aspects of sardines) feeding ecology and, as it is, the focus of the Introduction is restricted and then opened again, and this is not correct.

Considering the high number of food-items identified (over 15 000) I am not sure if the sample size is big enough to support the analysis. At least, a test on the minimum sample size must be included to substantiate the analysis.

As referred to above, this investigation is very descriptive and considerations related to  fisheries management are limited to a very general text in the last paragraph of the discussion.

Reviewer 3 Report

The manuscript adresses important topic - analyses of adult sardine Sardina pilchardus diet in the northernmost coastal area of the Mediterranean Sea. The authors present good information that improves the knowledge of the food web studies analysing sardine diet composition, assessing carbon uptake, estimating feeding selectivity. Such information is especially important for different modelling studies. Analysis is well done and clearly described. Very impressive is table with taxonomic composition of fish stomach composition. However to my opinion the main problem is that the authors present good information collected years 2006 and 2007 and it has good potential to be used in different modells for calibrations etc but they do not analyse if and how the diet of sardine change after these years. It could be done based on later research from other publications. In my opinion it substantionally would improve the value of manuscript.

Round 2

Reviewer 2 Report

I consider that the authors considered the comments and suggestions on the previous version of the Ms and made the adequate changes/adaptations. Thus, I consider that the Ms might be accepted in this revised format.